# Measuring exposure to misinformation from political elites on Twitter

Mohsen Mosleh [1,2,3] ✉ & David G. Rand [3,4,5]

Misinformation can come directly from public figures and organizations (referred to here as "elites"). Here, we develop a tool for measuring Twitter users' exposure to misinformation from elites based on the public figures and organizations they choose to follow. Using a database of professional fact-checks by PolitiFact, we calculate falsity scores for 816 elites based on the veracity of their statements. We then assign users an elite misinformation-exposure score based on the falsity scores of the elites they follow on Twitter. Users' misinformation-exposure scores are negatively correlated with the quality of news they share themselves, and positively correlated with estimated conservative ideology. Additionally, we analyze the co-follower, co-share, and co-retweet networks of 5000 Twitter users and observe an association between conservative ideology and misinformation exposure. Finally, we find that estimated ideological extremity is associated with more misinformation exposure to a greater extent for users estimated to be conservative than for users estimated to be liberal. Finally, we create an open-source R library and an Application Programming Interface (API) making our elite misinformation-exposure estimation tool openly available to the community.

There has been growing public concern about misinformation on social media. Accordingly, a great deal of effort has been invested by practitioners and researchers into investigating the spread of online misinformation[1,2]. Prior work has largely focused on belief in, and sharing of, articles (often headlines) from reliable versus unreliable news domains. While this kind of misinformation was a particular focus of interest in the wake of the 2016 election cycle (e.g., far-fetched headlines from little-known publishers going viral), a different form of misinformation has begun to gain attention in recent years: coordinated misinformation campaigns orchestrated by public figures and organizations (i.e., referred to here as "elites"), such as the claims of widespread fraud in the 2020 U.S. Presidential Election made by Republican politicians[3]. Given the large body of evidence documenting the impact of political elites on voter attitudes and behaviors[4-7], exploring exposure to elite-based misinformation (rather than just news domains) is important.

Furthermore, in focusing on what people believe and share, prior work has largely overlooked what (mis)information people are exposed to (a notable exception is ref. 8). Although exposure and sharing are obviously related (insomuch as you can only share content that you are exposed to), they are fundamentally different constructs. Most people share only a tiny fraction of the content they are exposed to[8], and therefore examining the content someone shares provides a very limited picture of a person's information environment. The choice of whom to follow (and thus what information to expose oneself to) is particularly important in light of evidence that simply being exposed to content, even if it is highly implausible, makes it subsequently seem truer[9].

Here, we introduce an approach for studying misinformation on social media that specifically focuses on exposure to misinformation from elites (defined as public figures and organizations). In particular, we estimate Twitter users' exposure to misinformation from elites by

[1]Management Department, University of Exeter Business School, Exeter, UK. [2]Alan Turing Institute, London, UK. [3]Sloan School of Management, Massachusetts Institute of Technology, Cambridge, MA, USA. [4]Initiative on the Digital Economy, Massachusetts Institute of Technology, Cambridge, MA, USA. [5]Department of Brain and Cognitive Sciences, Massachusetts Institute of Technology, Cambridge, MA, USA. ✉e-mail: m.mosleh@exeter.ac.uk

examining the extent to which they follow the accounts of elites who make false or inaccurate claims (based on PolitiFact ratings) to a greater or lesser degree. (We adapt an approach used in prior work for estimating social media users' partisanship by examining the ideological leanings of the political elites they follow[10], and apply that approach to misinformation.) The measure we introduce allows researchers to study users' choices about what level of (mis)information to expose themselves to and provides a tool for scholars of elite cues and messaging to examine exposure to elite misinformation online (e.g., to examine how this exposure correlates with other measures of interest). Our measure also allows researchers who study sharing to take a step towards controlling for exposure when estimating effects on sharing and provides an outcome measure for Twitter field experiments (e.g., that try to motivate users to improve the information environment they are exposed to).

Our approach also helps to address a methodological challenge facing social media studies of misinformation. These kinds of studies typically involve examining the content users post on social media[8,11,12], linking survey responses to social media data[13,14] or open web data[15], or conducting field experiments on social media and measuring the impact of interventions on subsequent sharing[16,17] (for methodological reviews, see refs. [18–20]). A key challenge for such studies is determining how to rate the quality of the content the users share or consume. The approach used in most prior studies is to assemble a list of domains with ratings: either a blacklist of domains classified as misinformation domains by journalists or fact-checkers, or continuous domain-level quality ratings generated by professional fact-checkers or crowd workers (e.g., refs. [21], [22]). These domain lists are then used to generate quality scores for each user (e.g., number of links to misinformation sites shared, average quality of links shared, etc.).

Although this domain-level approach has yielded many insightful results, there are important limitations. Nearly all prior work measures behavior that occurs after exposure—sharing[8,14,16,17] or clicking out to visit websites[15,23]. However, which content you are exposed to—and thus even have the chance to share or click on—is determined by the users you follow. Therefore, the choice of which accounts to follow has a profound impact on the information environment users experience online, but this choice has received much less attention from researchers in the study of misinformation. Additionally, there is a great deal of turnover among fake news sites (in part to evade social media platforms' efforts to block particularly egregious publishers). Thus, domain lists used in these approaches—which are incomplete to begin with—go stale fairly quickly, and there is no clear way to update the lists (e.g., no widely agreed-upon criteria for inclusion). And finally, by definition, the domain-based approach only captures posts with links—yet only a small minority of all posts contain links[17], and thus a substantial amount of potentially misleading content is missed.

The approach we introduce here addresses these limitations. We leverage a large public database of professional fact-checks by PolitiFact to generate falsity scores (a number between 0 and 1 based on the veracity of statements made by an elite, where 0 represents only entirely truthful statements and 1 represents only entirely false statements) for a range of elites (i.e., public figures and organizations). We then give each user a misinformation-exposure score by averaging the falsity scores associated with Twitter accounts of all the elites who the user choose to follow. Our approach measures the exposure of users to the content generated by the accounts they follow on Twitter, rather than relying on user behaviors. We use the accounts that a user follows as a proxy for what they would see in their feed (an approach which could easily be extended to platforms beyond Twitter). Additionally, our measure is based on a readily available public dataset of elites that is easily updated using fact-checking websites, and also easily updated to use sources of fact-checks beyond PolitiFact. Finally, our approach does not rely on identifying posts that include URLs. To make our measure broadly available to the research community, we

also created an open-source R library and an API that calculates Twitter users' misinformation-exposure scores.

We then validate our approach by showing that the resulting misinformation-exposure scores correlate with the quality of information users share online. We also identify communities of accounts and domains that are preferentially followed and shared by users with high misinformation exposure and observe an ideological asymmetry such that estimated political extremity is more associated with misinformation exposure for users estimated to be conservatives compared to liberals.

## Results

### Descriptives

We begin with descriptives of the fact-checking data we collected from PolitiFact. Figure 1a shows the distribution of the number of fact-checks per elite. Restricting to accounts with at least three fact-checks (and who are therefore included in our study), Fig. 1b shows the number of fact-checks per each category of rating, and Fig. 1c shows the distribution of falsity scores associated with each elite. Figure 1d shows the distribution of the elites' number of Twitter followers and Fig. 1e shows the number of elites followed by each Twitter user in our sample.

### Misinformation exposure and users' characteristics

Next, we look at the relationship between users' misinformation-exposure scores (as measured by averaging the falsity scores of the elite Twitter accounts they followed) and their characteristics, estimated from their digital fingerprints on Twitter. Consistent with our expectation that following more misinformation-spreading accounts (and thus being exposed to more misinformation) should result in sharing more misinformation oneself, users' misinformation-exposure scores are negatively correlated with the quality of content they shared. We measured news quality using domain-level trustworthiness ratings, generated in two different ways. First, we used the average rating of eight professional fact-checkers. Second, we used ratings collected from a sample of 970 Americans quota-matched to the national distribution on age, gender, ethnicity, and geographic region; the average rating of respondents who preferred the Democratic party were averaged with the ratings of the respondents who preferred the Republican party to create politically balanced layperson ratings. Users' misinformation-exposure score was negatively associated with the quality of information shared using both the professional fact-checker ratings (Fig. 2a; $b = -0.728$, 95% CI = [−0.753, −0.704], SE = 0.013, $t$ (3072) = −58.184, $p < 0.001$) and the politically balanced layperson ratings (Fig. 2b; $b = -0.540$, 95% CI = [−0.570,−0.510], SE = 0.015, $t$ (3072) = −35.299, $p < 0.001$).

Aligned with prior work finding that people who identify as conservative consume[15], believe[24], and share more misinformation[8,14,25], we also found a positive correlation between users' misinformation-exposure scores and the extent to which they are estimated to be conservative ideologically (Fig. 2c; $b = 0.747$, 95% CI = [0.727,0.767] SE = 0.010, $t$ (4332) = 73.855, $p < 0.001$), such that users estimated to be more conservative are more likely to follow the Twitter accounts of elites with higher fact-checking falsity scores. Critically, the relationship between misinformation-exposure score and quality of content shared is robust controlling for estimated ideology ($b = -0.712$, 95% CI = [−0.751, −0.673], SE = 0.020, $t$ (3067) = 36.008, $p < 0.001$ using professional fact-checker ratings; $b = -0.565$, 95% CI = [−0.613,−0.518], SE = 0.0124, $t$ (3067) = −23.387, $p < 0.001$ using crowd ratings), whereas the magnitude of the relationship between estimated ideology and quality of content is reduced when controlling for misinformation-exposure score ($b = -0.021$, 95% CI = [−0.058, 0.016], SE = 0.019, $t$ (3067) = −1.115, $p = 0.265$ using professional fact-checker ratings; $b = 0.030$, 95% CI = [−0.015, 0.076], SE = 0.023, $t$ (3067) = 1.307, $p = 0.191$ using crowd ratings) the coefficient of estimated ideology decreases by almost 100% (from −0.548 to −0.021 using professional fact-checker ratings and from

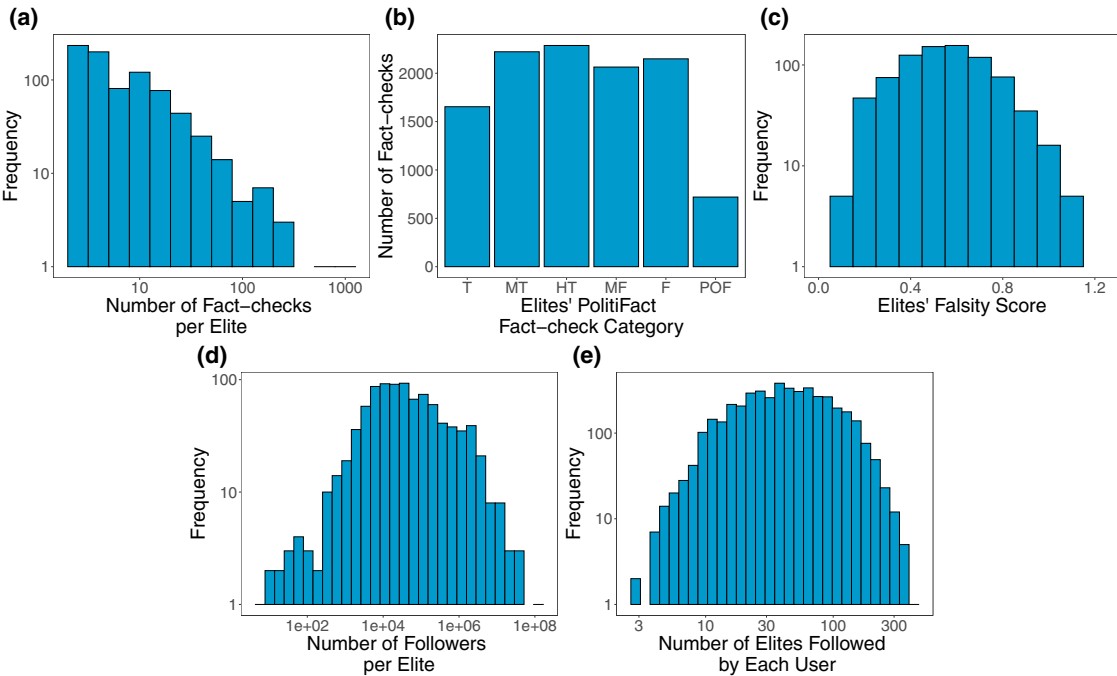

**Fig. 1 | Descriptives of the fact-checking dataset that forms the basis of our elite misinformation-exposure measure. a** Distribution of number of fact-checks per elite provided by PolitiFact. **b** Number of fact-checks per each PolitiFact category (T True, MT Mostly True, HT Half True, MF Mostly False, F False, POF Pants on Fire). **c** Distribution of falsity scores associated with each elite. **d** Distribution of number of Twitter followers of elites. **e** Distribution of number of elites followed by each Twitter user in our sample. Source data are provided as a Source Data file.

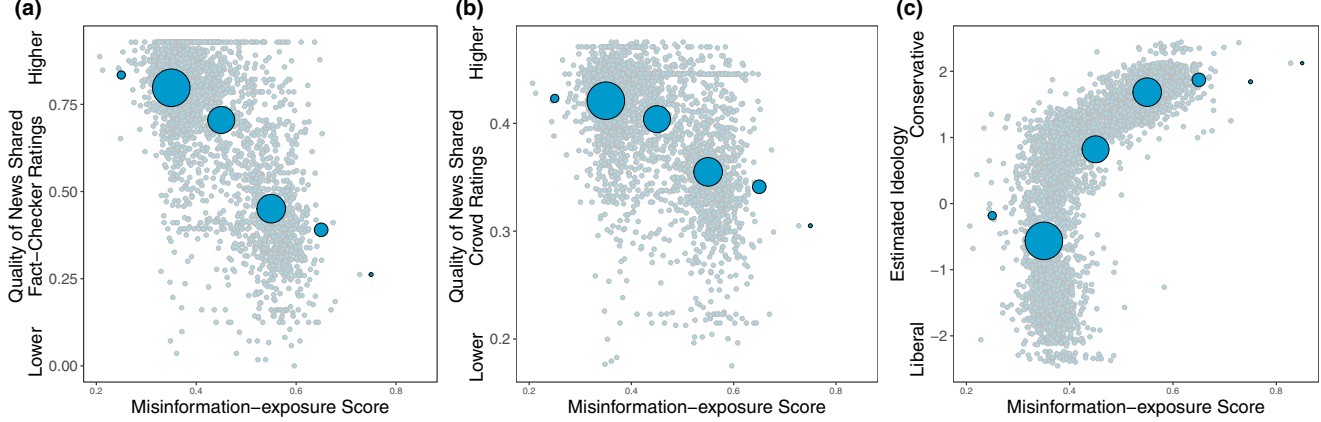

**Fig. 2 | Exposure to elite misinformation is associated with sharing news from lower-quality outlets and with conservative estimated ideology.** Shown is the relationship between users' misinformation-exposure scores and (**a**) the quality of the news outlets they shared content from, as rated by professional fact-checkers[21], (**b**) the quality of the news outlets they shared content from, as rated by layperson crowds[21], and (**c**) estimated political ideology, based on the ideology of the accounts they follow[10]. Small dots in the background show individual observations; large dots show the average value across bins of size 0.1, with size of dots proportional to the number of observations in each bin. Source data are provided as a Source Data file.

−0.388 to 0.030 using crowd ratings) and becomes insignificant ($p < 0.001$ to $p = 0.265$ using professional fact-checker ratings and $p < 0.001$ to $p = 0.191$ using crowd ratings) when we include misinformation exposure to predict quality of content. Thus, our misinformation-exposure score successfully isolates the predictive power of following inaccurate accounts (above and beyond estimated ideology), with misinformation-exposure explaining 53% of the variation in the quality of news sources shared when evaluating quality based on fact-checker ratings, and 29% of the variation in the quality of news sources shared when evaluating quality based on crowd ratings.

Given that toxicity and outrage may be associated with online misinformation[26], we also calculated the average language toxicity using Google Jigsaw Perspective API[27] and the average level of moral-outrage

language using a recently published estimator[28]. We found that misinformation-exposure scores are significantly positively related to language toxicity (Fig. 3a; $b = 0.129$, 95% CI = [0.098, 0.159], SE = 0.015, $t$ (4121) = 8.323, $p < 0.001$; $b = 0.319$, 95% CI = [0.274, 0.365], SE = 0.023, $t$ (4106) = 13.747, $p < 0.001$ when controlling for estimated ideology) and expressions of moral outrage (Fig. 3b; $b = 0.107$, 95% CI = [0.076, 0.137], SE = 0.015, $t$ (4143) = 14.243, $p < 0.001$; $b = 0.329$, 95% CI = [0.283, 0.374], SE = 0.023, $t$ (4128) = 14.243, $p < 0.001$ when controlling for estimated ideology). See Supplementary Tables 1, 2 for full regression tables and Supplementary Tables 3–6 for the robustness of our results.

We also note that the list of elites we used here can be used to estimate users' political ideology. To estimate ideology using the list of elites, we code the ideology of democrat elites as −1 and Republican

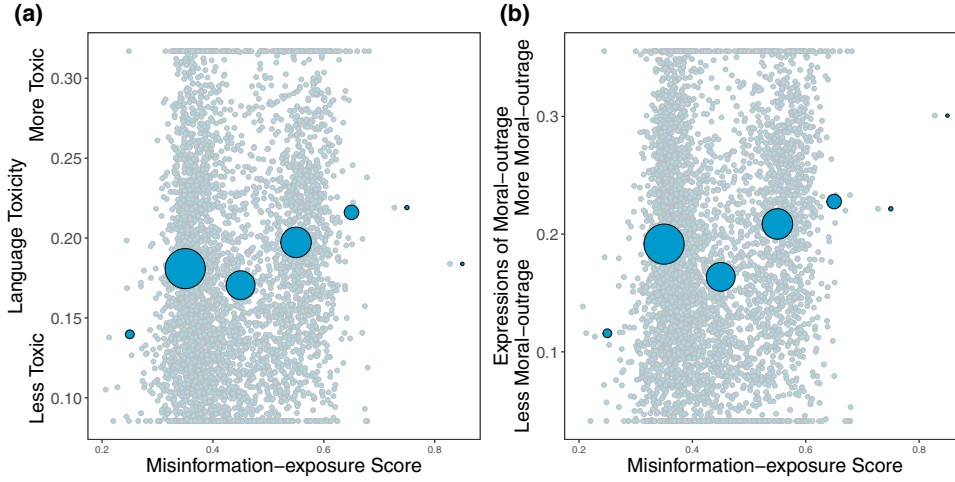

**Fig. 3 | Exposure to elite misinformation is associated with the use of toxic language and moral outrage.** Shown is the relationship between users' misinformation-exposure scores and (**a**) the toxicity of the language used in their tweets, measured using the Google Jigsaw Perspective API[27], and (**b**) the extent to which their tweets involved expressions of moral outrage, measured using the algorithm from ref. 28. Extreme values are winsorized by 95% quantile for visualization purposes. Small dots in the background show individual observations; large dots show the average value across bins of size 0.1, with size of dots proportional to the number of observations in each bin. Source data are provided as a Source Data file.

elites as 1, then for each user we average over the ideology of the elites they follow. Our measure of estimated ideology is strongly correlated with the follower-based ideology estimators of ref. 10 ($r = 0.86$, 95% CI = [0.855, 0.870], $t$ (4330) = 112.45, $p < 0.001$).

### Misinformation-exposure score and co-share network

Next, we gain more insight into the correlates of exposure to misinformation from elites by investigating which domains are preferentially shared by users with higher versus lower misinformation-exposure scores (for similar analyses of co-follower and co-retweet networks, see Supplementary Figs. 1, 2 and Supplementary Tables 7–10). To do so, we constructed a co-share network (see the Methods Section) of the 1798 domains that were shared by at least 20 users in our sample.

Community-detection analysis[29] on the co-share network reveals three distinct clusters of domains (Fig. 4a). Table 1 shows the 10 domains that are shared by the largest number of users in our sample in each cluster. The clusters differ in the average misinformation-exposure scores of users who shared them (Fig. 4b; average misinformation-exposure scores of users who shared domains in each cluster are cluster 1, 0.389; cluster 2, 0.404; cluster 3, 0.506), as well as their estimated ideology (Fig. 4c; average estimated ideology scores of each cluster are cluster 1, −0.470; cluster 2, 0.038; cluster 3, 1.22). Specifically, we see a cluster of domains estimated to be liberal, a cluster of center-left domains, and a cluster of domains estimated to be conservative, with misinformation exposure higher in the cluster of accounts estimated to be conservative compared to the other two clusters. Importantly, average misinformation-exposure scores of users who shared those domains differed significantly across clusters, even when controlling for average estimated ideology score ($p < 0.001$ with and without ideology control, Tukey's Honestly Significant Differences test).

Additionally, the clusters differ in the average toxicity of language use (Fig. 4d; average toxicity of language use of users who shared domains in each cluster are cluster 1, 0.186; cluster 2, 0.159; cluster 3, 0.199) and moral-outrage expressions (Fig. 4e; average moral-outrage expressions in each cluster are cluster 1, 0.213; cluster 2, 0.170; cluster 3, 0.226) of the users who shared the domains, such that the users of the more politically moderate cluster are less likely to use toxic and moral-outrage language compared to the clusters estimated to be liberal or conservative ($p < 0.001$ with and without ideology control, Tukey's Honestly Significant Differences test).

We found qualitatively similar results investigating the co-follower and co-retweet networks (see Supplementary Figs. 1, 2 and Supplementary Tables 7–10).

### Estimated ideological extremity and misinformation exposure

Finally, we complement the co-follower and co-share network analyses with a user-level analysis examining the relationship between estimated ideological extremity and misinformation exposure. To do so, we predict misinformation-exposure scores of the users in our sample using their estimated ideological extremity (i.e., absolute value of estimated ideology) interacted with binary estimated ideology (liberal versus conservative). We do this analysis using two different methods for robustness: (i) As in the rest of the paper, we use political ideology estimated from the political accounts users follow[10] and discretize based on scores above vs. below 0; and (ii) we estimate users' ideology based on their media sharing[30] and use the estimated ideology of the Associated Press (AP; a neutral outlet) as the cut-off (with users who share content that is on average more liberal than AP classified as liberal, and users who share content that is on average more conservative than AP classified as conservative).

We find that more ideologically extreme users are exposed to more misinformation—but, interestingly, this association is stronger among users estimated to be conservative compared to users estimated to be liberal. Specifically, we find a significant interaction between estimated conservative ideology and estimated ideological extremity (Fig. 5; $b = 0.756$, 95% CI = [0.726, 0.786], SE = 0.015, $t$ (4330) = 49.871, $p < 0.001$ when estimating ideology using accounts followed and $b = 0.415$, 95% CI = [0.367, 0.462], SE = 0.024, $t$ (3100) = 17.101, $p < 0.001$ when estimating ideology using news media sharing). Decomposing this interaction, we find a stronger association between estimated ideological extremity and misinformation exposure among users estimated to be conservative ($b = 0.825$, 95% CI = [0.804, 0.846], SE = 0.010, $t$ (2852) = 77.97, $p < 0.001$ when estimating ideology using accounts followed and $b = 0.567$, 95% CI = [0.523, 0.610], SE = 0.022, $t$ (1381) = 25.508, $p < 0.001$ when estimating ideology using news media sharing) than users estimated to be liberal ($b = 0.160$, 95% CI = [0.110, 0.211], SE = 0.025, $t$ (1478) = 6.255, $p < 0.001$ when estimating ideology using accounts followed and $b = 0.111$, 95% CI = [0.065, 0.159], SE = 0.023, $t$ (1719) = 4.659, $p < 0.001$ when estimating ideology using news media sharing). See Supplementary Table 11 for the full regression. We find

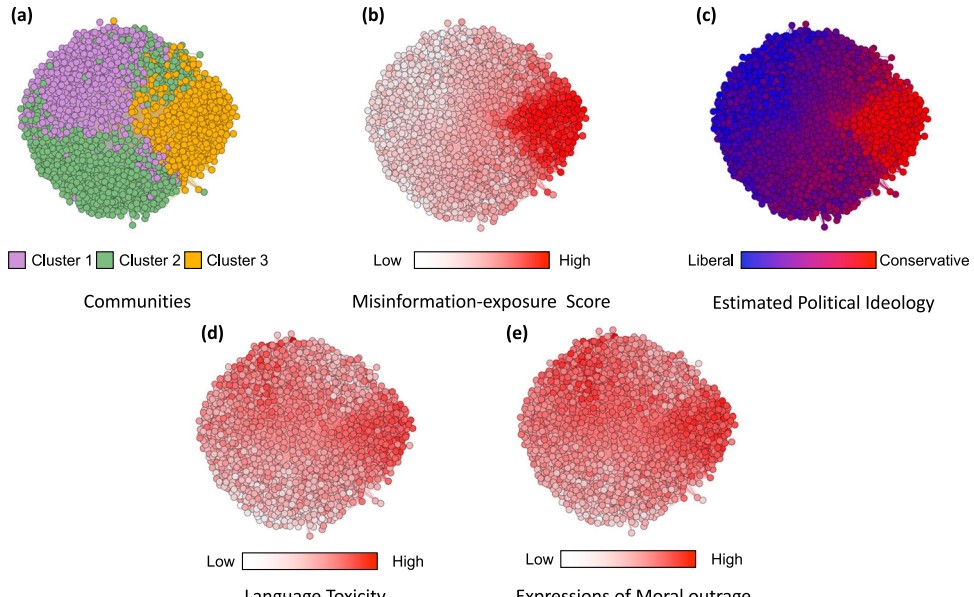

**Fig. 4 | In the co-share network, a cluster of websites shared more by conservatives is also shared more by users with higher misinformation exposure scores.** Nodes represent website domains shared by at least 20 users in our dataset and edges are weighted based on common users who shared them. **a** Separate colors represent different clusters of websites determined using community-detection algorithms[29]. **b** The intensity of the color of each node shows the average misinformation-exposure score of users who shared the website domain (darker = higher PolitiFact score). **c** Nodes' color represents the average estimated ideology of the users who shared the website domain (red: conservative, blue: liberal). **d** The intensity of the color of each node shows the average use of language toxicity by users who shared the website domain (darker = higher use of toxic language). **e** The intensity of the color of each node shows the average expression of moral outrage by users who shared the website domain (darker = higher expression of moral outrage). Nodes are positioned using directed-force layout on the weighted network.

a similar asymmetry when using language toxicity or moral outrage as the outcome, rather than misinformation-exposure score (see Fig. 6 and Supplementary Table 12).

## Discussion

In this work, we have introduced an approach for classifying Twitter users' exposure to misinformation from elites based on PolitiFact fact-checks of the accounts they choose to follow. We found that users who followed elites who made more false or inaccurate statements themselves shared news from lower-quality news outlets (as judged by both fact-checkers and politically-balanced crowds of laypeople), used more toxic language, and expressed more moral outrage. We also found that such users were more likely to be conservative, but that all of the previously mentioned associations were robust to controlling for estimated ideology. At the ecosystem level, we identified a cluster

## Table 1 | Top website domains in each cluster within the co-share network

| Cluster 1 | Cluster 2 | Cluster 3 |
| --- | --- | --- |
| nytimes.com | forbes.com | wsj.com |
| washingtonpost.com | apple.com | thehill.com |
| cnn.com | bbc.com | foxnews.com |
| politico.com | wordpress.com | pscp.tv |
| nbcnews.com | bbc.co.uk | nypost.com |
| go.com | espn.com | dailymail.co.uk |
| huffpost.com | change.org | washingtonexaminer.com |
| npr.org | vimeo.com | breitbart.com |
| yahoo.com | eventbrite.com | whitehouse.gov |
| cbsnews.com | twimg.com | senate.gov |

For each cluster, the table shows the 10 website domains in each cluster with the largest number of users who shared them in our sample in descending order. Source data are provided as a Source Data file.

of accounts that tended to be followed by, and domains that tended to be shared by, users who were estimated to be more conservative and who followed elites who made more false or inaccurate statements. And finally, at the individual level, we found that estimated ideological extremity was more strongly associated with following elites who made more false or inaccurate statements among users estimated to be conservatives compared to users estimated to be liberals. These results on political asymmetries are aligned with prior work on news-based misinformation sharing[31].

Our findings highlight the importance of the information people choose to expose themselves to when it comes to the spread of misinformation on social media. People who followed elites who made more false or inaccurate statements also shared news from lower-quality sources. This observation connects to research suggesting that political leaders' rhetoric can drive the beliefs and policy positions of their followers (rather than the leaders responding to the attitudes held by their constituents)[32–34]. It seems reasonable to hypothesize that following elites who make more false or inaccurate statements will cause citizens to believe more misinformation—future work should investigate this possibility. Furthermore, although misinformation exposure may sound like a passive process, in which users are incidentally or unintentionally exposed to misinformation, users on Twitter and other social media platforms have a substantial amount of control over what information they see. In particular, users in our study with high misinformation-exposure scores chose to follow accounts of elites that make false claims. The misinformation score we introduce here can thus be used in future work to examine predictors of which users select into following accounts of elites who make more false or inaccurate statements, as well as to distinguish the role of algorithmic recommendations versus individual user preferences.

Our analysis of co-follower and co-share networks suggests that the phenomenon of echo chambers, in which discourse is more likely with like-minded others, is not limited to politics. We also find evidence of "falsehood echo chambers", where users that are more often

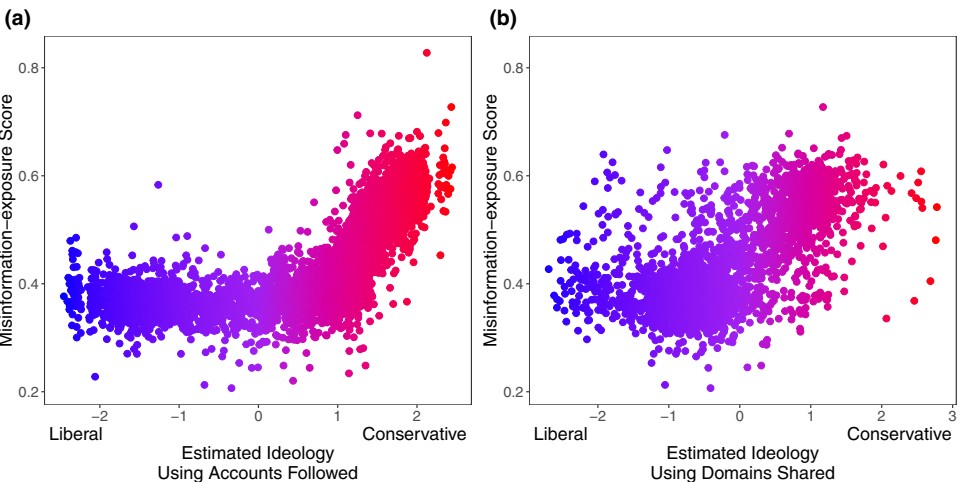

**Fig. 5 | Estimated ideological extremity is associated with higher elite misinformation-exposure scores for estimated conservatives more so than estimated liberals. a** Political ideology is estimated using accounts followed[10].

**b** Political ideology is estimated using domains shared[30] (Red: conservative, blue: liberal). Source data are provided as a Source Data file.

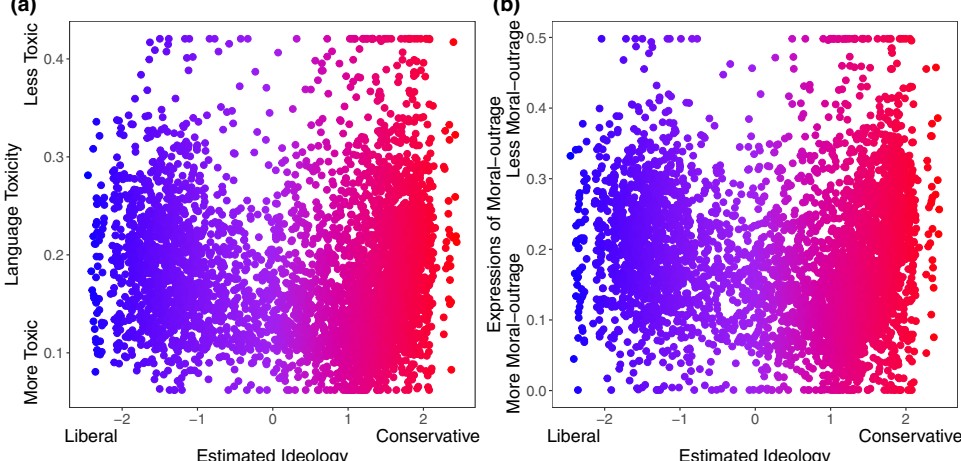

**Fig. 6 | Estimated ideological extremity is associated with higher language toxicity and moral outrage scores for estimated conservatives more so than estimated liberals.** The relationship between estimated political ideology and (**a**)

language toxicity and (**b**) expressions of moral outrage. Extreme values are winsorized by 95% quantile for visualization purposes. Source data are provided as a Source Data file.

exposed to misinformation are more likely to follow a similar set of accounts and share from a similar set of domains. These results are interesting in the context of evidence that political echo chambers are not prevalent, as typically imagined[35]. While the average American may be fairly politically apathetic and exposed to a broad spectrum of news[35,36], our results suggest that people who follow politicians on Twitter—and thus are likely to be more politically engaged than the average American—may indeed have more siloed exposure, when it comes to both partisan lean and level of misinformation. Furthermore, our results add to the growing body of literature documenting—at least at this historical moment—the link between extreme right-wing ideology and misinformation[8,14,24] (although, of course, factors other than ideology are also associated with misinformation sharing, such as polarization[25] and inattention[17,37]).

More generally, understanding the role of misinformation originating from cultural and political elites is of the utmost importance, given the influence that elites have on the attitudes of the masses. The approach we have introduced here offers a tool for exploring these issues across a range of applications. To that end, we have built an R package and an API that calculates misinformation-exposure scores and the number of rated elite accounts followed for any set of Twitter

users, and have posted the full set of falsity ratings and code online (both are publicly and freely available at https://github.com/mmosleh/minfo-exposure). When using this tool, researchers must decide how to handle the misinformation-exposure scores of users who follow only one or two rated accounts and may wish to weight their models by the precision of the misinformation-exposure score estimates. The tool we developed in this paper is designed to measure users' exposure to misinformation from political elites on Twitter. The tool can be used in scientific research for the purpose of quantifying the quality of content users experience on Twitter. Careful consideration of ethical and privacy issues should be considered in any future use of this research.

There are, of course, important limitations of the present work. Most notably, the misinformation-exposure score we calculate is based on the evaluations of the organization PolitiFact. Thus, to the extent that there is bias in which public figures and statements PolitiFact decides to fact-check, or in PolitiFact's evaluations, those biases will be carried forward into our ratings. For example, the partisan differences in falsity scores associated with elites Twitter accounts and misinformation exposure that we find could be explained, at least in part, by liberal bias on the part of PolitiFact in which claims they evaluate

and what conclusions they reach[38,39]. It is somewhat reassuring, however, that the misinformation-exposure scores we calculate are correlated with sharing links to low-quality news sites as judged not just by fact-checkers but also by politically balanced layperson crowds. Furthermore, the basic methodology we have introduced can be extended to use falsity ratings from any source, such as other fact-checking organizations (e.g., right-leaning sites such as TheDispatch.com) or crowd ratings (e.g., Twitter's Birdwatch program[40,41]). Another limitation of our approach is that users must follow a sufficient number of elite accounts in order to receive a misinformation-exposure score. Future work should investigate what user characteristics are predictive of following these elite accounts, and thus what forms of selection are induced by restricting to users who follow these elite accounts. Furthermore, our approach relies on users' immediate follower network and does not capture exposure beyond one's followed account (e.g., through retweets by users one is connected to, or by algorithmic recommendations). Relatedly, when assessing the association between exposure to elite misinformation and misinformation sharing, the measure of sharing quality that we used was able to assign quality ratings to only a small subset of links; future work should generalize these analyses to other misinformation-sharing metrics. Finally, our focus here is on Twitter, which is not a representative sample of the population; however, the approach we introduce here could be used on any other social media platform in which one can observe which pages or accounts a given user follows.

The approach we have introduced here opens new doors to understanding the role of exposure to misinformation from elites, and choices about who to follow online, in the spread of false and misleading claims over social media.

## Methods

### Data collection
We retrieved ratings for all 1005 elites (e.g., politicians, bureaucrats, famous personalities, advocacy groups, and media organizations) that, as of October 28, 2020, were fact-checked at least three times by the fact-checking website PolitiFact (restricting to those with at least four fact-checks gives similar results; see Supplementary Fig. 3). Each fact-check results in one of six ratings assigned by PolitiFact: True (the statement is accurate and there is nothing significant missing), Mostly True (the statement is accurate but needs clarification or additional information), Half True (the statement is partially accurate but leaves out important details or takes things out of context), Mostly False (the statement contains an element of truth but ignores critical facts that would give a different impression), False (the statement is not accurate), and Pants on Fire (the statement is not accurate and makes a ridiculous claim). For each elite, we then calculated a falsity score by assigning each fact-check a veracity rating using the scoring scheme from (True: 1, Mostly True: 0.8, Half True: 0.6, Mostly False: 0.4, False: 0.2, Pants on Fire: 0), averaging the rating of all fact-checks for that figure (creating an average veracity score), and then subtracting that score from 1 to convert from a veracity score to a falsity score. Our results are not unique to this particular scoring scheme (see Supplementary Table 4).

To use these falsity scores to rate Twitter users' exposure to misinformation from elites, we identified 950 Twitter accounts that are associated with 816 of the 1005 elites (we could not find Twitter accounts for the remaining 189 accounts; we found multiple Twitter accounts for some of the 816 and included all of them; our results are robust to excluding entities with multiple Twitter accounts and also excluding entities related to organizations; see Supplementary Table 5). We then collected all of the followers of these 950 accounts ($N = 122,562,681$ unique users). For assessment purposes, we created a list of users who followed at least three elites ($N = 38,328,679$) and then randomly sampled $N = 5000$ Twitter users from that list (see Supplementary Figs. 3, 4 for results when using alternative thresholds for

number of accounts followed). For each user in our sample, we calculated the misinformation-exposure score by averaging over the falsity scores of all rated elite accounts that the user followed, weighted by the average number of tweets per 2-month period in the past 2 years of the corresponding account as a proxy for intensity of exposure (our findings are robust when falsity scores are not weighted, see Supplementary Table 4; we could not retrieve the list of followed accounts for 650 of the 5000 users since they were protected accounts or did not exist anymore).

### Estimating users' political ideology
We estimated the political ideology of users based on accounts they followed[10]. The logic behind this approach is that users on social media are more likely to follow accounts that align with their own ideology than ideologically distant accounts. With the list of partisan elite accounts from ref. 10, we used their algorithm to calculate a continuous ideology score on the interval $[-2.5, 2.5]$ for each user, where $-2.5$ represents a strong liberal ideology and 2.5 represents a strong conservative ideology. We used ideology midpoint 0 to classify users as conservative versus liberal.

### Quantifying quality of content
We quantified the quality of content shared by each user using a list of domains for which we have professional fact-checkers' trustworthiness ratings, similar to prior work[16,17]. We did so by collecting each user's last 3200 tweets on July 23, 2021 (capped by the Twitter API limit; we could not retrieve the timeline of 837 out of 5000 users since they were protected accounts, did not have any tweets, or did not exist anymore), identifying all tweets that contained URLs, and averaging the trustworthiness ratings of the news domains each URL linked to. Specifically, we used a list of 60 news websites[21] (20 fake news, 20 hyper-partisan, and 20 mainstream news websites), each of whose trustworthiness had been rated [0, 1] by eight professional fact-checkers, as well as by a large politically balanced crowd of demographically representative (quota-sampled) laypeople living in the United States recruited via Lucid (the crowd was balanced in terms of its representation of Democrats and Republicans). Of 5,363,779 total links shared by users in our sample, 5% were from the list of news websites for which we had fact-checkers' ratings. For each user, we calculated the quality of content shared by averaging the rated news websites they had shared.

### Constructing co-share network
Following prior work on cognitive echo chambers on social media[13], we constructed a co-share network where nodes represent domains that are shared by at least 20 users in our sample and weighted edges represent the number of users who mutually shared them. For each domain, we averaged over estimated ideology, misinformation-exposure scores, language toxicity, and expression of moral outrage of users who shared that website. We used a similar approach to construct co-follower (see Supplementary Fig. 1; Supplementary Tables 7, 8) and co-retweet networks (see Supplementary Fig. 2; Supplementary Tables 9, 10).

All statistics reported in the results are conducted using linear regression models with standardized coefficients, unless otherwise stated. All statistical tests are two-tailed. All data were collected using Twitter API and Python. Our study received a waiver from an ethics review by the MIT Committee on the Use of Humans as Experimental Subjects (COUHES) protocol E-3973. In conducting our study, Twitter and PolitiFact data were obtained in line with the Terms of Service of the websites.

### Reporting summary
Further information on research design is available in the Nature Portfolio Reporting Summary linked to this article.

## Data availability

PolitiFact data are available in ref. 42. Twitter data contains identifiable information–and for confidentiality reasons–are only available upon request. The Twitter data are available under restricted access for research purposes, access can be obtained by writing to the authors. Source data are provided with this paper.

## Code availability

All code used to generate the results are available on https://osf.io/5283b/[43]. An R package and an API that calculates misinformation-exposure scores is publicly available at https://github.com/mmosleh/minfo-exposure.

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

## Acknowledgements

The authors thank Jennifer O'Conner for her assistance with finding Twitter accounts of PolitiFact elites, Barrett Golding for providing the elite fact-check data from PolitiFact, Brian Guay and Ben Tappin for insightful comments, and Gregory Eady for assistance in estimating political ideology using media sharing. The authors gratefully acknowledge funding from Google through a Google Research Scholar Award (MM), as well as funding from the Alfred P. Sloan Foundation Grant #2021-16891, the National Science Foundation FAIN 2047152, and the TDF Foundation, (DGR). The authors acknowledge support from Amazon Web Services for providing AWS computing credits (MM).

## Author contributions

M.M. and D.G.R. designed the research. MM accessed PolitiFact data and Twitter data. M.M. conducted the analysis. M.M. and D.G.R. wrote the paper.

## Competing interests

M.M. and D.R. have received research funding from Google, and D.R. has received research funding from Meta.
