## [Peer Review File · Nature Communications]

Measuring exposure to misinformation from political elites on TwitterREVIEWER COMMENTS

Reviewer #1 (Remarks to the Author):

In this paper the authors introduce a novel method for calculating the exposure of Twitter users to misinformation from political elites. They use ratings from the well-respected professional fact-checking website Politifact to determine how frequently each of a large set of public figures lies, and then calculate misinformation exposure for any Twitter user by averaging the lie frequency of the public figures they follow. The authors then use this measure to demonstrate that conservative echo chambers contain more lying public figures than liberal echo chambers, and also show an important asymmetry between Democrats and Republicans whereby partisan extremism is associated with greater exposure to elite misinformation for Republicans but not Democrats.

I believe that this is an extremely important paper about a topic of great importance. The paper generates novel insights into elite misinformation, a topic which has not received nearly as much attention as it deserves. Furthermore, the method introduced here is very powerful, and the authors have produced an open access R library implementing their algorithm. I am confident that many researchers will adopt this tool, and the paper will become highly cited.

Therefore, I overall strongly support publication of the paper in Nature Communications. That being said, I believe there are some revision that would be helpful.

First, it would instructive to extend the network analysis – the authors currently look at co-follow networks for Twitter accounts (as well as co-share networks for news domains). They should complement the co-follow analysis with a co-retweet network analysis. For example, many Democrats might follow Trump but do not retweet (endorse) from him. So it would be important to create a network of Twitter accounts where connections represent the number of users who retweeted both accounts, and replicate the same analyses done for the co-follow and co-share networks. If the results are similar, that adds important robustness. If the results are different, that sheds interesting new light.

Second, I found the results on moral outrage and language toxicity to be interesting, but somewhat conceptually disconnected from the other results. The authors should do more to motivate why they look at these constructs; and also give substantially more detail on what exactly the moral-outrage classifier is (as this is a new classifier – I had not seen it prior to this work). What exactly does it represent and how should it be interpreted?

Finally, the results on asymmetries seems somewhat related to Kikolov et al 2021 <http://doi.org/10.37016/mr-2020-55> looking at news (rather than elite) misinformation – it would be useful to mention this resonance in the discussion.

Reviewer #2 (Remarks to the Author):

This is a well-executed manuscript on a novel topic. All too often, misinformation is posed as a problem emanating from how individual users respond to fake news articles they encounter. Yet such encounters are actually rare. Seemingly more frequent, and certainly neglected by the literature, is exposure to misinformation propagated directly by political elites, usually through social networks like Twitter. The authors of the present paper rectify this gap, and offer interesting data on the relationship between the penchant for falsehoods of political leaders whom users choose to follow, and characteristics of those users.

I think this article should be published. Below, I outline several questions and concerns meant to improve the manuscript:

-How exactly were the Twitter users randomly sampled?

-How did you deal with political leaders for whom there were multiple Twitter accounts? Do the results change at all when you remove those leaders?

-Can figure 1 be amended to account for # of followers per public figure? It would be great to know how falsity corresponds with popularity

-Figure 2 needs to be re-done. There's not enough information about each sub-figure to make sense of them (e.g., I don't know how to interpret the ideology results or the toxicity scores). At minimum, add a lengthier caption. I'd definitely re-do the labels if I were you. These graphs should communicate results without having to read the rest of the paper.

-I don't like the blobs in Figures 2 and 3. Is there any way you can just show individual observations, perhaps with some jittering?

-You write: "Thus, our misinformation-exposure score successfully isolates the role following untrustworthy accounts plays in the sharing of misinformation (above and beyond ideology)." Can you provide a precise measure of this? How much of sharing misinfo is attributable to following untrustworthy accounts?

-Do you worry at all that your community detection approach yields an otherwise neutral cluster with Bill Clinton? I'm just not sure that has face validity. Given the computational complexity of the task, it's possible something erroneous happened through no fault of your own. If I were you, I'd go back and re-run the procedures that produced that result. If everything holds, please explain why Clinton would appear in this cluster.

-The subsection "Misinformation-exposure score and co-share network" should begin by explaining what question it is trying to answer. Right now, it seems entirely redundant to the section which preceded it.

-There are two broader questions I'd like you to consider, at least in the discussion section. First, what do your results imply for our understanding of the relationship between political leaders' rhetoric and their followers? There is a vast literature on this topic (e.g., Lenz 2012), sometimes with application to misinformation (e.g., Nyhan and Reifler's "The Effect of Fact-Checking on Elites: A Field Experiment on U.S. State Legislators"). Second, what do your results imply for our understanding of echo chambers? Recent work in this area (e.g., Guess's "(Almost) Everything in Moderation: New Evidence on Americans' Online Media Diets" and Becker et al's "Wisdom of Partisan Crowds") has indicated that concerns about echo chambers may be overblown. Your findings would seem to complicate that consensus.

-The writing throughout the paper needs revisions. For example:

-rewrite for missing word: "We adapt an approach used in prior work for estimating social media users' partisanship by examining the ideological leanings of the political elites they [8], and apply that approach to misinformation.)

-this entire paragraph needs to be re-written: "We find a significant interaction between conservative ideology and ideological extremity..." As it stands, I have trouble sussing out what that paragraph is trying to communicate.

-missing a close parens in this sentence: "To that end, we have built an R package"

Finally, and perhaps this is just personal taste, but the Github README with the R package should list what R version is required and whether there are any dependencies. I was able to easily install it on one of my machines, less easily on another. On a related note, you should convey the scale of the

misinfo score when misinfo.score is printed.

All in all, however, this is a compelling article that I look forward to seeing in print.

Reviewer #3 (Remarks to the Author):

Thank you for the chance to review "Falsehood in, falsehood out: Measuring exposure to elite misinformation on Twitter." Overall, I'm somewhat conflicted about this piece. On the one hand, this paper addresses the problem of elite misinformation, which has been critically understudied in this subfield. However, there are significant limitations here (reliance on PolitiFact, e.g.). Still, this project may represent the best we can do, as a subfield, at this moment. I do have a number of more specific comments and concerns, discussed below.

First, the authors write that understanding elite-based misinformation is important in light of the literature's focus on e.g., link sharing from dubious domains. I certainly agree. However, might it be useful to more directly compare these? I realize that this paper tests and finds a correlation between elite exposure scores and quality of Twitter users' own news sharing. It seems that there is quite a bit of overlap between the degree of exposure/consumption implied by these two approaches – is that a correct assessment? If so, that would also be a valuable conclusion of this analysis. The authors could additionally emphasize how these two "scores" differ. What, if anything, would we miss by focusing on either list-based or crowdsourced evaluations of news sharing from these users, rather than their elite-misinformation-exposure score?

Doing so could also help highlight the implications of the paper. Currently, the authors stress that elite misinfo is important, and understudied. However, I think the authors can do more to explain what is useful about this approach in improving our understanding of misinformation. I commend the authors on making their tool available – but why should researchers use this, exactly? What theoretical questions can it answer beyond the current toolkit? What can it shed light on? For example, the authors write on p 16 that "The misinformation score we introduce here can be used in future work to examine predictors of which users select into following less trustworthy elite accounts, as well as to distinguish the role of algorithmic recommendations versus individual user preferences." These implications can be emphasized earlier in the paper. The authors can also do more explain how their findings extend our theoretical understanding of misinformation (though I want to be clear that I am not asking for "novel theory" here).

A key methodological issue I see is the inclusion of media orgs in the list of elites in the elite exposure score calculation. The authors stress that their approach complements studies of fake "news" domain link sharing by focusing on political elites, who likely account for a greater share of misinformation according to some observers. However, they subsequently measure "elite" exposure by scraping the full Politifact list that includes not only political elites and advocacy groups, but many of those same dubious/fake "news" domains (as well as mainstream news orgs). There seems to be a conflation, then. What do the results look like when news orgs and websites are removed and only actual political elites are included?

Of course, the authors note that the reliance on Politifact might be a limitation, but I believe it is a more severe one than they apparently do, as there is not much engagement with how this might bias their inferences. The authors write that they find misinformation exposure scores are associated with conservatism, and that ideological extremity is association with misinformation exposure for conservatives but not liberals. However, is it not possible that these patterns are artefacts of the use of Politifact? Some research has noted potential partisan disparities in selection processes of PolitiFact. For example, "[a] content analysis of the published fact-checks addressing three disputed realities ... suggests substantial differences in the questions asked and the answers offered, limiting the

usefulness of fact-checking for citizens trying to decide which version of disputed realities to believe," (Marietta et al 2015) while others find that "PolitiFact was more likely to find greater deceit in Republican rhetoric." (Farnsworth & Lichter, 2016).

Marietta, M., Barker, D. C., & Bowser, T. (2015, December). Fact-checking polarized politics: Does the fact-check industry provide consistent guidance on disputed realities?. In *The Forum* (Vol. 13, No. 4, pp. 577-596). De Gruyter.

Farnsworth, S. J., & Lichter, S. R. (2016, September). A comparative analysis of the partisan targets of media fact-checking: examining President Obama and the 113th Congress. In American Political Science Association Annual Meeting, Philadelphia, PA, September.

Finally, I have a number of smaller-scale questions about the study or its limitations.

- There are a number of assumptions in the creation of the exposure score. The authors use the PF score for a proxy of what followers see, for instance, although the twitter feed is not directly mapped to what is fact-checked, obviously.
- Why is weighting for tweet frequency not the primary analysis (only supplemental) – if the interest is in exposure to misinformation, frequency should be key.
- The analysis focuses on accounts followed to assess exposure; what about retweets and quote tweets? – e.g. virtually everyone saw many of President Trump's tweets despite not following him and were thus exposed to misinfo. In fact, controversial and dubious tweets from public figures are probably especially likely to gain outsized attention and penetrate outside their follower network.
- Obviously, Twitter is a smallish slice of the population, disproportionately made up of engaged political hobbyists. How might we better address broader exposure?
- The authors used the trustworthiness ratings for a list of 60 domains (20 msm, 20 fake, 20 hyperpartisan) from Pennycook & Rand, 2019 to assess the trustworthiness of news shared by the twitter sample. What % of links were covered by this list? Which if any domains were not rated? I'm not clear on how this step of the analysis proceeded.

Reviewer #4 (Remarks to the Author):

This paper provides a new method of assessing misinformation exposure on Twitter. Building upon a gold-standard method of inferring political orientation in online environments, the authors develop and validate a misinformation-exposure score, share their open-source tool with the public and showcase its use in misinformation research. This is an excellent manuscript, bearing an important contribution to the field of misinformation research, and will be of interest to the readership of *Nature Communications*. I recommend accepting the manuscript pending minor revision.

1. I want to commend the authors on their contribution. The paper details a rigorous development of the misinformation-exposure metric. It then validates it on several other metrics, notably language associated with toxicity and moral outrage. I have already stated I recommend accepting the paper pending minor revision. Please see my comments below.

2. I take issue with the scoring scheme. I understand that the results are robust compared to other scoring schemes, but I would like the authors to expand, even in the SI, on how this scoring scheme was developed and what negative values mean. I bring this up because the index is easily understood in the [0,1] range. Some public figures receive negative values, which is unintuitive. In addition, why has a scoring scheme of [0,1] in increments of 1/6 not been considered?

3. A related issue to point (2): I have downloaded and fiddled with the R package. In the current version, falsity and veracity scores are the same. The conversion from veracity to falsity does not apply. I assume this is a problem with the public R package and not with the reported analysis in this manuscript, but please double-check.

4. Whereas the analysis on page 6 is sound (misinformation exposure and quality of content), it took a

few readings to interpret. I would recommend the authors would add full regression tables to the SI.

5. As mentioned, the fact that conservatives share more misinformation is consistent with prior findings. However, it should be noted that this is consistent with findings that take other theoretical frameworks (e.g., Osmundsen, Bor, Vahlstrup, Bechmann & Petersen, 2021; APSR). I would advise the authors to credit opposing but consistent perspectives as well.

6. To my understanding, on p.11 is, the co-share network is about users who *share* information from a list of domains. If so, why are they being referred to as "followers" and not "sharers": "use of their followers, such that the followers of the politically neutral cluster being less likely to use toxic and moral-outrage language compared to the liberal or conservative cluster"

REVIEWER COMMENTS

Reviewer #1 (Remarks to the Author):

In this paper the authors introduce a novel method for calculating the exposure of Twitter users to misinformation from political elites. They use ratings from the well-respected professional fact-checking website Politifact to determine how frequently each of a large set of public figures lies, and then calculate misinformation exposure for any Twitter user by averaging the lie frequency of the public figures they follow. The authors then use this measure to demonstrate that conservative echo chambers contain more lying public figures than liberal echo chambers, and also show an important asymmetry between Democrats and Republicans whereby partisan extremism is associated with greater exposure to elite misinformation for Republicans but not Democrats.

I believe that this is an extremely important paper about a topic of great importance. The paper generates novel insights into elite misinformation, a topic which has not received nearly as much attention as it deserves. Furthermore, the method introduced here is very powerful, and the authors have produced an open access R library implementing their algorithm. I am confident that many researchers will adopt this tool, and the paper will become highly cited.

Therefore, I overall strongly support publication of the paper in Nature Communications. That being said, I believe there are some revision that would be helpful.

Thank you very much for this positive evaluation, and for your helpful comments!

First, it would instructive to extend the network analysis – the authors currently look at co-follow networks for Twitter accounts (as well as co-share networks for news domains). They should complement the co-follow analysis with a co-retweet network analysis. For example, many Democrats might follow Trump but do not retweet (endorse) from him. So it would be important to create a network of Twitter accounts where connections represent the number of users who retweeted both accounts, and replicate the same analyses done for the co-follow and co-share networks. If the results are similar, that adds important robustness. If the results are different, that sheds interesting new light.

Thank you for the great suggestion. We have now added additional analysis of the co-retweet network in the SI and show our results are qualitative the same. Interestingly we found that the co-retweet network is more polarized compared to co-follower network, suggesting that retweets are a stronger signal of one's preferences.

Supplementary Figure 2 Co-retweet network. Nodes represent Twitter accounts retweeted by at least 20 users in our dataset and edges are weighted based on retweeted accounts. (a) Separate colors represent different clusters using community detection algorithms [1]. (b) The intensity of color of each node shows the average misinformation-exposure score of its followers (darker = higher misinformation-exposure score). (c) Nodes' color represents the average ideology of the followers (red: conservative, blue: liberal). (d) Average language toxicity used by the followers (darker = more toxic language use). (e) Average moral-outrage expressions (darker = more expressions of moral-outrage). Nodes are positioned using directed-force layout on the weighted network. See Supplementary Table 8 for average characteristics of users who retweeted accounts in each cluster and Supplementary Table 9 for top 10 accounts in each cluster.

Second, I found the results on moral outrage and language toxicity to be interesting, but somewhat conceptually disconnected from the other results. The authors should do more to motivate why they look at these constructs; and also give substantially more detail on what exactly the moral-outrage classifier is (as this is a new classifier – I had not seen it prior to this work). What exactly does it represent and how should it be interpreted?

We have added text describing the important role that outrage and toxicity play in the spread of misinformation, to better motivate why we include those constructs. We have also clarified that the classifier we use was developed as part of a recently published paper in Science Advances by Brady et al.

“Given that toxicity and outrage are often associated with online misinformation²³, we also calculated the average language toxicity using Google Jigsaw Perspective API²⁴ and average level of moral-outrage language using a recently published classifier²⁵.”

Finally, the results on asymmetries seems somewhat related to Kikolov et al 2021 <http://doi.org/10.37016/mr-2020-55> looking at news (rather than elite) misinformation – it would be useful to mention this resonance in the discussion.

Thank you for your suggestion we have now included citation to that paper in the discussion Section:

“And finally, at the individual level, we found that ideological extremity was much more strongly associated with following less truthful elites for users estimated as conservatives than users estimated as liberals. These results on political asymmetries are aligned with prior work on news-based misinformation sharing [27].”

Reviewer #2 (Remarks to the Author):

This is a well-executed manuscript on a novel topic. All too often, misinformation is posed as a problem emanating from how individual users respond to fake news articles they encounter. Yet such encounters are actually rare. Seemingly more frequent, and certainly neglected by the literature, is exposure to misinformation propagated directly by political elites, usually through social networks like Twitter. The authors of the present paper rectify this gap, and offer interesting data on the relationship between the penchant for falsehoods of political leaders whom users choose to follow, and characteristics of those users.

I think this article should be published. Below, I outline several questions and concerns meant to improve the manuscript:

Thank you very much for this positive evaluation, and your very helpful comments!

-How exactly were the Twitter users randomly sampled?

We created a list of users who followed at least three of the public figures for whom we had politifact ratings ($N= 38,328,679$) and then randomly sampled $N=5,000$ Twitter users from that list. We have now clarified this in the Method section.

“... We then collected all of the followers of these 950 accounts (Total $N=122,562,681$). For assessment purposes, we created a list of users who followed at

least three elites (N=38,328,679) and then randomly sampled N=5,000 Twitter users from that list”

-How did you deal with political leaders for whom there were multiple Twitter accounts? Do the results change at all when you remove those leaders?

Our main analyses aggregate across all accounts we could find for each politician (i.e. users who follow any of a given politician’s accounts are counted as following that politician). As per your suggestion, we have now included a robustness check in the SI Table 5 that excludes elites with multiple accounts.

*Supplementary Table 5 Robustness of results when including only Twitter accounts related to public political figures (excluding accounts related to organizations) and when excluding elites with multiple Twitter accounts. ($p < .1$, $*p < 0.05$, $**p < 0.01$, $***p < 0.001$)*

	Dependent variable	Independent variable	Not Excluding anyone	Excluding organizations' Twitter accounts	Excluding elites with multiple Twitter accounts
Model 1	Fact-checkers ratings	Intercept	-0.037** (0.012)	-0.038** (0.013)	-0.036** (0.013)
		Misinformation-exposure score	-0.728*** (0.013)	-0.71*** (0.013)	-0.721*** (0.013)
Model 2	Crowd ratings	Intercept	-0.027 (0.015)	-0.028 (0.015)	-0.027 (0.015)
		Misinformation-exposure score	-0.54*** (0.015)	-0.519*** (0.015)	-0.533*** (0.016)
Model 3	Fact-checkers ratings	Intercept	-0.039** (0.013)	-0.04** (0.013)	-0.04** (0.013)
		Misinformation-exposure score	-0.712*** (0.02)	-0.689*** (0.021)	-0.684*** (0.02)
		Political ideology	-0.021 (0.019)	-0.026 (0.02)	-0.049 (0.019)
Model 4	Crowd ratings	Intercept	-0.026 (0.015)	-0.028 (0.016)	-0.027 (0.015)
		Misinformation-exposure score	-0.565*** (0.024)	-0.531*** (0.025)	-0.539*** (0.024)
		Political ideology	0.03 (0.023)	0.014 (0.024)	0.006 (0.023)
Model 5	Political ideology	Intercept	-0.001 (0.01)	-0.001 (0.01)	0 (0.01)
		Misinformation-exposure score	0.746*** (0.01)	0.759*** (0.01)	0.737*** (0.01)

-Can figure 1 be amended to account for # of followers per public figure? It would be great to know how falsity corresponds with popularity

Great suggestion, we have included the distribution of followers per public figures as another panel (d) in Figure 1. We find a weak (marginally significant) correlation between log-transformed of number of followers (we used log-transform since the value is highly skewed) and falsity score ($r=-.062, p=.060$).

Figure 1 (a) Distribution of number of fact-checks per elite provided by PolitiFact. (b) Number of fact-checks per each PolitiFact category (T: True, MT: Mostly True, HT: Half True, MF: Mostly False, F: False, POF: Pants on Fire). (c) Distribution of elites' falsity scores. (d) Distribution of number of followers of elites. (e) Distribution of number of elites followed by each user.

-Figure 2 needs to be re-done. There's not enough information about each sub-figure to make sense of them (e.g., I don't know how to interpret the ideology results or the toxicity scores). At minimum, add a lengthier caption. I'd definitely re-do the labels if I were you. These graphs should communicate results without having to read the rest of the paper.

We have added more details to the axis labels and figures legends for better communication as follows:

Figure 2. The relationship between users' misinformation-exposure scores and (a) quality of news outlets they shared content from, as rated by professional fact-checkers²¹, (b) quality of news outlets they shared content from, as rated by politically-balanced layperson crowds²¹, and (c) estimated political ideology, based on the ideology of the accounts they follow¹⁰. Small dots in the background show individual observations; large dots shows the average value across bins of size of 0.1, with size of dots proportional to the number of observations in each bin.

Figure 3. The relationship between users' misinformation-exposure scores and (a) the toxicity of the language used in their tweets, measured using the Google Jigsaw Perspective API²⁴ and (b) the extent to which their tweets involved expressions of moral outrage, measured using the algorithm from ref 25. Extreme values are winsorized by 95%-quantile for visualization purpose. Small dots in the background show individual observations; large dots shows the average value across bins of size of 0.1, with size of dots proportional to the number of observations in each bin.

-I don't like the blobs in Figures 2 and 3. Is there any way you can just show individual observations, perhaps with some jittering?

Great suggestion, We have updated Figs 2 and 3 to include individual observations. See previous comment.

-You write: "Thus, our misinformation-exposure score successfully isolates the role following untrustworthy accounts plays in the sharing of misinformation (above and beyond ideology)." Can you provide a precise measure of this? How much of sharing misinfo is attributable to following untrustworthy accounts?

Thank you for this suggestion! In the revised text, we use the r2 of the models to quantify this question, and write:

"Thus, our misinformation-exposure score successfully isolates the role following untrustworthy accounts plays in predicting the sharing of misinformation (above and beyond estimated ideology); with misinformation-exposure explaining 53% of the

variation in the quality of news sources shared when evaluating quality based on fact-checker ratings, and 29% of the variation in the quality of news sources shared when evaluating quality based on crowd ratings.”

-Do you worry at all that your community detection approach yields an otherwise neutral cluster with Bill Clinton? I'm just not sure that has face validity. Given the computational complexity of the task, it's possible something erroneous happened through no fault of your own. If I were you, I'd go back and re-run the procedures that produced that result. If everything holds, please explain why Clinton would appear in this cluster.

Thank you for the comment. We have checked the procedure and can confirm that results for Cluster 2 are not due to an error. After reflecting on the contents of the cluster, we have re-labeled it as “center left” rather than “neutral”.

-The subsection "Misinformation-exposure score and co-share network" should begin by explaining what question it is trying to answer. Right now, it seems entirely redundant to the section which preceded it.

Thank you for your comment. We have now added an opening section to that section. We have also moved the co-follower network and co-retweet network (added per Reviewer 1 comment) to the SI

“Misinformation-exposure score and co-share network

Next, we gain more insight into the correlates of misinformation exposure by investigating which domains are preferentially shared by users with higher versus lower misinformation-exposure scores (for similar analysis on co-follower and co-retweet network see See Supplementary Figure 1,2 and Supplementary Table 6-9). To do so, we constructed a co-share network (see the Method Section) of the 1,798 domains that were shared by at least 20 users in our sample.”

-There are two broader questions I'd like you to consider, at least in the discussion section. First, what do your results imply for our understanding of the relationship between political leaders' rhetoric and their followers? There is a vast literature on this topic (e.g., Lenz 2012), sometimes with application to misinformation (e.g., Nyhan and Reifler's "The Effect of Fact-Checking on Elites: A Field Experiment on U.S. State Legislators").

Thank you for making this connection. In our revised discussion, we now speak to this point as follows:

“People who followed more dishonest elites also themselves shared news from lower quality sources. This observation connects to research suggesting that political leaders’ rhetoric can drive the beliefs and policy positions of their followers (rather than the leaders responding to the attitudes held by their constituents)³¹⁻³³. It seems reasonable to hypothesize that following dishonest elites will cause citizens to believe more misinformation – future work should investigate this possibility.”

Second, what do your results imply for our understanding of echo chambers? Recent work in this area (e.g., Guess’s “(Almost) Everything in Moderation: New Evidence on Americans’ Online Media Diets” and Becker et al’s “Wisdom of Partisan Crowds”) has indicated that concerns about echo chambers may be overblown. Your findings would seem to complicate that consensus.

This is an interesting point! We have added the following to the discussion:

These results are interesting in the context of evidence that political echo-chambers are not prevalent at typically imagined³¹. While the average American may be fairly politically apathetic and exposed to a broad spectrum of news, our results suggest that people who follow politicians on Twitter – and thus are likely to be more politically engaged than the average American – may indeed have more siloed exposure, when it comes to both partisan lean and level of misinformation.

-The writing throughout the paper needs revisions. For example:

-rewrite for missing word: "We adapt an approach used in prior work for estimating social media users’ partisanship by examining the ideological leanings of the political elites they [8], and apply that approach to misinformation.)

We have fixed the missing word in the main text, and hired a professional copy-editor to proof read our manuscript.

-this entire paragraph needs to be re-written: "We find a significant interaction between conservative ideology and ideological extremity... " As it stands, I have trouble sussing out what that paragraph is trying to communicate.

We have rewritten the paragraph as follows:

We find that more ideologically extreme users are exposed to more misinformation – but, interestingly, this association is much stronger among conservative users compared to liberal users. Specifically, we find a significant interaction between estimated conservative ideology and ideological extremity (Figure 5; $p < 0.001$ when estimating ideology using accounts followed or news media sharing). Decomposing this

interaction, we find a much stronger association between ideological extremity and misinformation-exposure among conservative users ($b=0.825$ or $b=0.567$, depending on ideology measure) than liberal users ($b=0.160$ or $b=0.111$, depending on ideology measure; all $p<0.001$). See Supplementary Table 11 for full regression table. We find a similar asymmetry when using language toxicity or moral outrage as the outcome, rather than misinformation exposure score (see Figure 6 and Supplementary Table 12).

-missing a close parens in this sentence: "To that end, we have built an R package"

We have fixed this in the main text

Finally, and perhaps this is just personal taste, but the Github ReadME with the R package should list what R version is required and whether there are any dependencies. I was able to easily install it on one of my machines, less easily on another. On a related note, you should convey the scale of the misinfo score when misinfo.score is printed.

We have included this information on GitHub

All in all, however, this is a compelling article that I look forward to seeing in print.

Many thanks for your time to review our paper!

Reviewer #3 (Remarks to the Author):

Thank you for the chance to review "Falsehood in, falsehood out: Measuring exposure to elite misinformation on Twitter." Overall, I'm somewhat conflicted about this piece. On the one hand, this paper addresses the problem of elite misinformation, which has been critically understudied in this subfield. However, there are significant limitations here (reliance on PolitiFact, e.g.). Still, this project may represent the best we can do, as a subfield, at this moment. I do have a number of more specific comments and concerns, discussed below.

Thank you for your very helpful comments and suggestions, which we have addressed as described below.

First, the authors write that understanding elite-based misinformation is important in light of the literature's focus on e.g., link sharing from dubious domains. I certainly agree. However, might it be useful to more directly compare these? I realize that this paper tests and finds a correlation between elite exposure scores and quality of Twitter users' own news sharing. It seems that there is quite a bit of overlap between the degree of exposure/consumption implied by these two

approaches – is that a correct assessment? If so, that would also be a valuable conclusion of this analysis. The authors could additionally emphasize how these two “scores” differ. What, if anything, would we miss by focusing on either list-based or crowdsourced evaluations of news sharing from these users, rather than their elite-misinformation-exposure score? Doing so could also help highlight the implications of the paper. Currently, the authors stress that elite misinfo is important, and understudied. However, I think the authors can do more to explain what is useful about this approach in improving our understanding of misinformation. I commend the authors on making their tool available – but why should researchers use this, exactly? What theoretical questions can it answer beyond the current toolkit? What can it shed light on? For example, the authors write on p 16 that “The misinformation score we introduce here can be used in future work to examine predictors of which users select into following less trustworthy elite accounts, as well as to distinguish the role of algorithmic recommendations versus individual user preferences.” These implications can be emphasized earlier in the paper. The authors can also do more explain how their findings extend our theoretical understanding of misinformation (though I want to be clear that I am not asking for “novel theory” here).

Thank you for raising this important point. From a theoretical perspective, although exposure and sharing are obviously related (in so much as you can only share content that you are exposed to), they are fundamentally different constructs. Most people share only a tiny fraction of the content they are exposed to, and therefore simply studying sharing gives a very limited picture of the information environment in which the person operates. The choice of who to follow (and thus what information to expose yourself to) is particularly important in light of evidence that simply being exposed to content, even if it is highly implausible, makes the content subsequently seem more true. Our measure allows researchers to study users’ choices about what level of (mis)information they want to be exposed to. Our measure also allows researchers who do study sharing to take a step towards accounting for exposure when estimating effects on sharing. Furthermore, by specifically focusing on exposure to misinformation from elites, our measure allows the large number of scholars who study elite cues/messaging to examine the phenomenon of online exposure to elite misinformation, for example by correlating users’ misinformation exposure score with outcomes relevant to their particular questions of interest, or by using the misinformation exposure score measure as an outcome for field experiments on Twitter (e.g. that try to motivate users to improve the information environment they are exposed to). In the revised introduction, we now discuss the theoretical distinction between exposure and sharing and value of studying exposure (2nd paragraph), and the potential uses of an elite misinformation exposure measure (3rd paragraph):

“Furthermore, in focusing on what people believe and share, prior work has largely overlooked what (mis)information people are exposed to (a notable exception is ref ⁸). Although exposure and sharing are obviously related (in so much as you can only

share content that you are exposed to), they are fundamentally different constructs. Most people share only a tiny fraction of the content they are exposed to⁸, and therefore examining the content someone shares provides a very limited picture of a person's information environment. The choice of whom to follow (and thus what information to expose oneself to) is particularly important in light of evidence that simply being exposed to content, even if it is highly implausible, makes the content subsequently seem more true⁹.

Here, we introduce a new approach for studying misinformation on social media that specifically focuses on exposure to misinformation from elites (defined as public figures and organizations). In particular, we estimate Twitter users' exposure to misinformation from elites by examining the extent to which they follow the account of elites who lie (based on Politifact ratings) to a greater or lesser degree. (We adapt an approach used in prior work for estimating social media users' partisanship by examining the ideological leanings of the political elites they follow⁸, and apply that approach to misinformation.) The measure we introduce allows researchers to study users' choices about what level of (mis)information to expose themselves to, and provides a new tool for scholars of elite cues and messaging to examine exposure to elite misinformation online (e.g. to examine how this exposure correlates with other measures of interest). Our measure also allows researchers who study sharing to take a step towards controlling for exposure when estimating effects on sharing, and provides a new outcome measure for Twitter field experiments (e.g. that try to motivate users to improve the information environment they are exposed to)."

Additionally, we have added empirical results in Supplementary Information Table 6 showing that associations between misinformation exposure score and users characteristics are significantly different from the association between link-based quality measure and users characteristics

"Empirically, the associations between misinformation exposure score and users characteristics are significantly different from the association between link-based quality measure and users characteristics. For example, misinformation exposure score is more correlated with estimated political ideology compared to link-based quality measure (almost 50% stronger correlation than for quality of content shared). Furthermore, misinformation exposure score is less correlated with language toxicity and expressions of moral-outrage compared to the correlation with quality of content shared (correlation with quality of content shared and language toxicity and is almost 2 times stronger; see Supplementary Information Table 6)."

Supplementary Table 6 Correlation of quality of content shared and misinformation exposures score with users' characteristics. All correlations are significant ($p < 0.001$). The magnitude of correlations for misinformation exposure score are significantly different from those for fact-checkers ratings and crowd ratings ($p < 0.001$ using confidence intervals to compare correlations; [3]).

	Estimated political ideology	Use of toxic language	Expressions of moral outrage
Misinformation exposure score	0.75	0.13	0.11
Misinformation Sharing (fact-checker ratings)	-0.57	-0.25	-0.25
Misinformation Sharing (crowd ratings)	-0.40	-0.19	-0.20

A key methodological issue I see is the inclusion of media orgs in the list of elites in the elite exposure score calculation. The authors stress that their approach complements studies of fake “news” domain link sharing by focusing on political elites, who likely account for a greater share of misinformation according to some observers. However, they subsequently measure “elite” exposure by scraping the full Politifact list that includes not only political elites and advocacy groups, but many of those same dubious/fake “news” domains (as well as mainstream news orgs). There seems to be a conflation, then. What do the results look like when news orgs and websites are removed and only actual political elites are included?

This is a good point. We have now included robustness checks of our results when we excluded news organizations from our list of rated Twitter accounts in the SI, and the results are virtually identical as shown in the following. Only 12% of the rated elite Twitter accounts are related to news organizations, which have 2.5% of the total followers.

Supplementary Table 5 Robustness of results when including only Twitter accounts related to public political figures (excluding accounts related to organizations) and when excluding elites with multiple Twitter accounts. ($p < .1$, $*p < 0.05$, $**p < 0.01$, $***p < 0.001$)

	Dependent variable	Independent variable	Not Excluding anyone	Excluding organizations' Twitter accounts	Excluding elites with multiple Twitter accounts
Model 1	Fact-checkers ratings	Intercept	-0.037** (0.012)	-0.038** (0.013)	-0.036** (0.013)
		Misinformation-exposure score	-0.728*** (0.013)	-0.71*** (0.013)	-0.721*** (0.013)
Model 2	Crowd ratings	Intercept	-0.027 (0.015)	-0.028 (0.015)	-0.027 (0.015)
		Misinformation-exposure score	-0.54*** (0.015)	-0.519*** (0.015)	-0.533*** (0.016)
Model 3	Fact-checkers ratings	Intercept	-0.039** (0.013)	-0.04** (0.013)	-0.04** (0.013)
		Misinformation-exposure score	-0.712*** (0.02)	-0.689*** (0.021)	-0.684*** (0.02)
		Political ideology	-0.021 (0.019)	-0.026 (0.02)	-0.049 (0.019)
Model 4	Crowd ratings	Intercept	-0.026 (0.015)	-0.028 (0.016)	-0.027 (0.015)
		Misinformation-exposure score	-0.565*** (0.024)	-0.531*** (0.025)	-0.539*** (0.024)
		Political ideology	0.03 (0.023)	0.014 (0.024)	0.006 (0.023)
Model 5	Political ideology	Intercept	-0.001 (0.01)	-0.001 (0.01)	0 (0.01)
		Misinformation-exposure score	0.746*** (0.01)	0.759*** (0.01)	0.737*** (0.01)

Supplementary Table 5 Robustness of results when including only Twitter accounts related to public political figures (excluding accounts related to organizations) and when excluding elites with multiple Twitter accounts. ($p < .1$, $*p < 0.05$, $**p < 0.01$, $***p < 0.001$)

	Dependent variable	Independent variable	Not Excluding anyone	Excluding news organizations' Twitter accounts	Excluding elites with multiple Twitter accounts
Model 1	Fact-checkers ratings	Intercept	-0.037** (0.012)	-0.038** (0.013)	-0.036** (0.013)
		Misinformation-exposure score	-0.728*** (0.013)	-0.71*** (0.013)	-0.721*** (0.013)
Model 2	Crowd ratings	Intercept	-0.027 (0.015)	-0.028 (0.015)	-0.027 (0.015)
		Misinformation-exposure score	-0.54*** (0.015)	-0.519*** (0.015)	-0.533*** (0.016)
Model 3	Fact-checkers ratings	Intercept	-0.039** (0.013)	-0.04** (0.013)	-0.04** (0.013)
		Misinformation-exposure score	-0.712*** (0.02)	-0.689*** (0.021)	-0.684*** (0.02)
		Political ideology	-0.021 (0.019)	-0.026 (0.02)	-0.049 (0.019)
Model 4	Crowd ratings	Intercept	-0.026 (0.015)	-0.028 (0.016)	-0.027 (0.015)
		Misinformation-exposure score	-0.565*** (0.024)	-0.531*** (0.025)	-0.539*** (0.024)
		Political ideology	0.03 (0.023)	0.014 (0.024)	0.006 (0.023)
Model 5	Political ideology	Intercept	-0.001 (0.01)	-0.001 (0.01)	0 (0.01)
		Misinformation-exposure score	0.746*** (0.01)	0.759*** (0.01)	0.737*** (0.01)

Of course, the authors note that the reliance on Politifact might be a limitation, but I believe it is a more severe one than they apparently do, as there is not much engagement with how this might bias their inferences. The authors write that they find misinformation exposure scores are associated with conservatism, and that ideological extremity is association with misinformation exposure for conservatives but not liberals. However, is it not possible that these patterns are artefacts of the use of Politifact? Some research has noted potential partisan disparities in selection processes of PolitiFact. For example, “[a] content analysis of the published fact-checks addressing three disputed realities ... suggests substantial differences in the questions asked and the answers offered, limiting the usefulness of fact-checking for citizens trying to decide which version of disputed realities to believe,” (Marietta et al 2015) while others find that “PolitiFact was more likely to find greater deceit in Republican rhetoric.” (Farnsworth & Lichter, 2016).

This is an important point, and we have revised the limitations section of the Discussion to move explicitly address this:

“Most notably, the misinformation-exposure score we calculate is based on the evaluations of the organization PolitiFact. Thus, to the extent that there is bias in which public figures and statements PolitiFact decides to fact-check, or in PolitiFact’s evaluations, those biases will be carried forward into our ratings. For example, the partisan differences in falsity scores and misinformation exposure that we find could be explained, at least in part, by liberal bias on the part of PolitiFact in which claims they evaluate and what conclusions they reach^{33,34}. It is somewhat reassuring, however, that the misinformation exposure scores we calculate are strongly correlated with sharing links to low quality news sites as judged not just by fact-checkers, but also by politically-balanced layperson crowds. Furthermore, the basic methodology we have introduced can be extended to use falsity ratings from any source, such as other fact-checking organizations (e.g. right-leaning sites such as TheDispatch.com) or crowd ratings (e.g. Twitter’s Birdwatch program³⁵).”

Finally, I have a number of smaller-scale questions about the study or its limitations.

- There are a number of assumptions in the creation on the exposure score. The authors use the PF score for a proxy of what followers see, for instance, although the twitter feed is not directly mapped to what is fact-checked, obviously.

This is an important point and we have clarified in the text that we used the accounts a user follows as a proxy for what they would see in their feed.

“... . We use the accounts a user follows as a proxy for what they would see in their feed.”

- Why is weighting for tweet frequency not the primary analysis (only supplemental) – if the interest is in exposure to misinformation, frequency should be key.

Thank you for your suggestion. We have revised the paper to use the weighted scoring as the primary analysis used in the main text, and moved the unweighted version to the SI robustness check section.

- The analysis focuses on accounts followed to assess exposure; what about retweets and quote tweets? – e.g. virtually everyone saw many of President Trump’s tweets despite not following him and were thus exposed to misinfo. In fact, controversial and dubious tweets from public

figures are probably especially likely to gain outsized attention and penetrate outside their follower network.

This is an important limitation of our work, which we now acknowledge in the Discussion:

“Furthermore, our approach relies on users’ immediate follower network, and does not capture exposure beyond ones’ followed account (e.g., through retweets by users one is connected to, or by algorithmic recommendations).”

- Obviously, Twitter is a smallish slice of the population, disproportionately made up of engaged political hobbyists. How might we better address broader exposure?

This is a good question. The basic approach we propose could be used on any other social media platform in which one can observe which pages or accounts a given user follows. We have now mentioned this in the discussion section.

- The authors used the trustworthiness ratings for a list of 60 domains (20 msm, 20 fake, 20 hyperpartisan) from Pennycook & Rand, 2019 to assess the trustworthiness of news shared by the twitter sample. What % of links were covered by this list? Which if any domains were not rated? I’m not clear on how this step of the analysis proceeded.

We have provided more details in the method section and mentioned the percentage of links covered by our list of rated domains.

“Of 5,363,779 total links shared by users in our sample, 5% were from the list of news websites for which we had fact-checkers ratings on interval [0,1]. For each user, we calculated the quality of content shared by averaging rated news websites they had shared.”

We have also added a note about this coverage issue to the limitations section:

“Relatedly, when assessing the association between exposure to elite misinformation and misinformation sharing, the measure of sharing quality that we used was able to assign quality ratings to only a small subset of links; future work should generalize these analyses to other misinformation sharing metrics.”

Reviewer #4 (Remarks to the Author):

This paper provides a new method of assessing misinformation exposure on Twitter. Building upon a gold-standard method of inferring political orientation in online environments, the

authors develop and validate a misinformation-exposure score, share their open-source tool with the public and showcase its use in misinformation research. This is an excellent manuscript, bearing an important contribution to the field of misinformation research, and will be of interest to the readership of Nature Communications. I recommend accepting the manuscript pending minor revision.

1. I want to commend the authors on their contribution. The paper details a rigorous development of the misinformation-exposure metric. It then validates it on several other metrics, notably language associated with toxicity and moral outrage. I have already stated I recommend accepting the paper pending minor revision. Please see my comments below.

Thank you very much for this positive evaluation, and your very helpful comments!

2. I take issue with the scoring scheme. I understand that the results are robust compared to other scoring schemes, but I would like the authors to expand, even in the SI, on how this scoring scheme was developed and what negative values mean. I bring this up because the index is easily understood in the $[0,1]$ range. Some public figures receive negative values, which is unintuitive. In addition, why has a scoring scheme of $[0,1]$ in increments of $1/6$ not been considered?

Thank you for your suggestion. We have now updated our analysis and used a scoring scheme of $[0,1]$ throughout the paper.

3. A related issue to point (2): I have downloaded and fiddled with the R package. In the current version, falsity and veracity scores are the same. The conversion from veracity to falsity does not apply. I assume this is a problem with the public R package and not with the reported analysis in this manuscript, but please double-check.

Thank you for your comment. We have fixed the issues with the GitHub repository.

4. Whereas the analysis on page 6 is sound (misinformation exposure and quality of content), it took a few readings to interpret. I would recommend the authors would add full regression tables to the SI.

Thank you for your suggestion. We have included full regression tables in the SI as follows

Supplementary Table 1 Regression models predicting misinformation-exposure score using fact-checkers ratings and crowd's ratings of content users shares and their estimated political ideology. ($p < .1$, $*p < 0.05$, $**p < 0.01$, $***p < 0.001$)

	Dependent variable	Independent variable	
Model 1	Fact-checkers ratings	Intercept	-0.037** (0.012)
		Misinformation-exposure score	-0.728*** (0.013)
Model 2	Crowd ratings	Intercept	-0.027 (0.015)
		Misinformation-exposure score	-0.54*** (0.015)
Model 3	Fact-checkers ratings	Intercept	-0.039** (0.013)
		Misinformation-exposure score	-0.712*** (0.02)
		Political ideology	-0.021 (0.019)
Model 4	Crowd ratings	Intercept	-0.026 (0.015)
		Misinformation-exposure score	-0.565*** (0.024)
		Political ideology	0.03 (0.023)
Model 5	Political ideology	Intercept	-0.001 (0.01)
		Misinformation-exposure score	0.746*** 0.01

5. As mentioned, the fact that conservatives share more misinformation is consistent with prior findings. However, it should be noted that this is consistent with findings that take other theoretical frameworks (e.g., Osmundsen, Bor, Vahlstrup, Bechmann & Petersen, 2021; APSR). I would advise the authors to credit opposing but consistent perspectives as well.

Thank you for your suggestion. When we discuss the prior literature on conservatives sharing more misinformation, we now also mention other theoretical frameworks beyond ideology for explaining misinformation sharing, such as polarization and inattention (including a cited to the suggestion Osmundsen et al. paper)

6. To my understanding, on p.11 is, the co-share network is about users who *share* information from a list of domains. If so, why are they being referred to as “followers” and not “sharers”:

“use of their followers, such that the followers of the politically neutral cluster being less likely to use toxic and moral-outrage language compared to the liberal or conservative cluster”

Thank you for pointing this out. We have fixed this typo in the text.

REVIEWERS' COMMENTS

Reviewer #1 (Remarks to the Author):

The authors have revised their manuscript comprehensively and with love to detail. I warmly recommend publication in Nature Communications in present form.

Reviewer #2 (Remarks to the Author):

This is an extremely well-done revision. You have responded effectively to the other reviewers' comments, and to my comments as well. The new figures represent substantial improvements; the language is more clear; and the re-naming of the cluster with Clinton to "center-left" makes a lot of sense. I support publication. That said, I do have a few comments:

-You write: "Decomposing this interaction, we find a much stronger association between ideological extremity and misinformation exposure among conservative users ($b=0.825$ or $b=0.567$, depending on ideology measure) than liberal users ($b=0.160$ or $b=0.111$, depending on ideology measure; all $p<0.001$)." To be clear, the difference is significant at $p<0.001$ --right? If not, I would change the language so as not to imply a statistical comparison. For similar reasons, I was also interested in seeing more details about this sentence: "Interestingly, we found the co-retweet network to be more polarized compared to the other networks, suggesting retweeting is a stronger signal of one's preferences." What's the difference, and what's the p-value? The authors should review all reported comparisons in the paper to make sure that they are comprehensively and precisely described.

-Typo. You write: "We did so by collecting their last 3,200 tweets on July 23, 2021 (capped by the Twitter API limit; we could not retrieve the timeline of 837 out of 5,000 users since they were protected accounts, did have any tweet, or did not exist anymore)..." In the revision, "did not have any tweet" should be "did not have any tweets"

-The inclusion of individual observations in Figure 2 makes me much more confident in several of your conclusions, especially about misinformation exposure and ideology. Figure 3, however, left me with more questions than answers. As your appendix tables show, the relationship is significant, but the eye test is going to leave readers confused. If I were you, I'd relegate Figure 3 to the appendix. (Substantively, it's not adding a lot to your story anyways.)

Again, this is a well-done paper that has only become better with revision.

Reviewer #3 (Remarks to the Author):

Thanks for the opportunity to review this revision. The authors have done a commendable job addressing my concerns. This manuscript tackles a timely issue with rigorous methods, and in its current iteration acknowledges its important, but not fatal, limitations.

Reviewer #4 (Remarks to the Author):

I would like to congratulate the authors on their significant contribution. My concerns have been addressed and I have no further comments.

REVIEWERS' COMMENTS

Reviewer #1 (Remarks to the Author):

The authors have revised their manuscript comprehensively and with love to detail. I warmly recommend publication in Nature Communications in present form.

We highly appreciate the reviewer's comments and time they took to review our paper

Reviewer #2 (Remarks to the Author):

This is an extremely well-done revision. You have responded effectively to the other reviewers' comments, and to my comments as well. The new figures represent substantial improvements; the language is more clear; and the re-naming of the cluster with Clinton to "center-left" makes a lot of sense. I support publication. That said, I do have a few comments:

We are thankful to the reviewer's comments. Please see below for our responses.

-You write: "Decomposing this interaction, we find a much stronger association between ideological extremity and misinformation exposure among conservative users ($b=0.825$ or $b=0.567$, depending on ideology measure) than liberal users ($b=0.160$ or $b=0.111$, depending on ideology measure; all $p<0.001$)." To be clear, the difference is significant at $p<0.001$ --right? If not, I would change the language so as not to imply a statistical comparison.

Yes, the difference is significant (as captured by the interaction reported in the preceding sentence).

For similar reasons, I was also interested in seeing more details about this sentence:

"Interestingly, we found the co-retweet network to be more polarized compared to the other networks, suggesting retweeting is a stronger signal of one's preferences." What's the difference, and what's the p-value? The authors should review all reported comparisons in the paper to make sure that they are comprehensively and precisely described.

Thank you for your comment, we have removed that sentence (we believe the details are beyond the scope of the current paper)

-Typo. You write: "We did so by collecting their last 3,200 tweets on July 23, 2021 (capped by the Twitter API limit; we could not retrieve the timeline of 837 out of 5,000 users since they were protected accounts, did have any tweet, or did not exist anymore)..." In the revision, "did not have any tweet" should be "did not have any tweets"

Thank you! We fixed the typo

-The inclusion of individual observations in Figure 2 makes me much more confident in several of your conclusions, especially about misinformation exposure and ideology. Figure 3, however,

left me with more questions than answers. As your appendix tables show, the relationship is significant, but the eye test is going to leave readers confused. If I were you, I'd relegate Figure 3 to the appendix. (Substantively, it's not adding a lot to your story anyways.)

While respecting the reviewers comment, we would like to keep that figure in the main text for transparency.

Again, this is a well-done paper that has only become better with revision.

Many thanks for your time reviewing our paper!

Reviewer #3 (Remarks to the Author):

Thanks for the opportunity to review this revision. The authors have done a commendable job addressing my concerns. This manuscript tackles a timely issue with rigorous methods, and in its current iteration acknowledges its important, but not fatal, limitations.

Thank you for the time you took in reviewing our paper

Reviewer #4 (Remarks to the Author):

I would like to congratulate the authors on their significant contribution. My concerns have been addressed and I have no further comments.

Thank you for the time you took in reviewing our paper